# Molecular Biomarkers for Early Detection of Alzheimer’s Disease and the Complementary Role of Engineered Nanomaterials: A Systematic Review

**DOI:** 10.3390/ijms26199282

**Published:** 2025-09-23

**Authors:** Muhammad Zia Ul Haq, Xinyi Zhao, Samuel Obeng Apori, Baljit Singh, Furong Tian

**Affiliations:** 1Nanolab Research Centre, Physical to Life Sciences Research Hub, Technological University Dublin, Camden Row, D08 CKP1 Dublin, Ireland; ziaulhaqm893@gmail.com (M.Z.U.H.); d20127084@mytudublin.ie (X.Z.); d21125192@mytudublin.ie (S.O.A.); baljit.singh@tudublin.ie (B.S.); 2School of Food Science and Environmental Health, Technological University Dublin, Grangegorman, D07 ADY7 Dublin, Ireland; 3MiCRA Biodiagnostics Technology Gateway, and Health, Engineering & Materials Science Research Hub, Technological University Dublin, D24 FKT9 Dublin, Ireland

**Keywords:** Alzheimer’s disease, biomarkers, neurological, nanoparticles, analytical methods, diagnostics, limit of detection, population distribution

## Abstract

Alzheimer’s disease (AD) instantly requires affordable diagnostic tools for targeting the responsible molecular biomarkers. In this review, we briefly discussed the overview of the AD population, performance of different analytical techniques and nanoparticles/composites, molecular biomarkers, and the interest of countries towards the detection of AD biomarkers during 2012–2025. The desired result was attained by lateral flow assay, surface-enhanced Raman scattering, and colorimetric sensor techniques with nanoparticles of magnetic, gold, and carbon-containing silver, and iridium oxide nanoparticles, upon biomarkers of dopamine, amyloid beta41, and Apolipoprotein E, individually. Additionally, the outstanding performance of nanoparticles including gold nanoparticles, carbon-containing nanoparticles, and manganese dioxide with their particle size of 5.7 nm, 35 nm, 37.3 nm, 120 nm, and 220 nm, respectively, has been discussed. Moreover, the percentages of AD-related biomarkers including amyloid beta42 having research articles of 21.2%, amyloid beta1-42 12.1%, amyloid beta oligomer 12.1%, phosphorylated Tau detection 12.1%, amyloid beta1-40 9.09%, Dopamine 9.09%, amyloid beta40 9.17%, apolipoprotein 6.06%, etc., have also been included. Additionally, LOD comparison with respect to applied analytical techniques, investigated through a timeline and electrochemical sensor, was found most suitable. Finally, a portable molecular diagnostic device to combine amyloid beta1-42, amyloid beta1-40, and phosphorylated Tau detection in non-invasive bodily fluid was proposed for the future and clinical diagnosis.

## 1. Introduction

Alzheimer’s disease (AD) is specially focused on in the current neurological and biomedical research. AD affects millions globally, with a significant societal and economic burden. The global societal cost of dementia was ~US$1.3 trillion in 2019. One of the most pressing challenges in Alzheimer’s research is early detection to start treatment on time to reduce patient symptoms and reduce long-term care costs. Current diagnostic delays often lead to missed opportunities for therapeutic intervention. This tool empowers clinicians to make timely decisions, improving patient outcomes [1,2].

The prevalent cause of dementia is AD, which affects the central nervous system (CNS) by accumulating β-amyloid plaques and intraneuronal neurofibrillary tau tangles (NFTs). Chronic AD mostly affects adults aged 65 years and older, and a total of 57 million people are affected globally. In terms of population health, dementia accounted for ~44 million disability-adjusted life years (DALYs) in 2019, and 36.3 million DALYs in 2021. The basic etiology of dementia is still unknown, but it is today a major problem in terms of prevalence and societal implications due to its obvious link with the aging process of the population, lifestyle, environmental factors, and genetic variables [3,4,5,6]. AD is a degenerative disease that mainly causes the loss of memory and death of neurons [7]. Furthermore, aberrant protein accumulation in the brain (tau tangles and β-amyloid plaques) causes brain cell death and a progressive deterioration in cognitive abilities. Non-cognitive changes include mood disorders, agitation, wandering, psychosis, altered personality, poor judgment, and irregular sleep habits.

Generally, there are three stages for the development of AD: (1) pre-symptomatic AD, (2) mild cognitive impairment (MCI) and (3) gradual progression into full-blown AD. Although AD symptoms might vary, younger people are more likely to experience atypical forms, including logogenic aphasia, behavioral variant AD, posterior cortical atrophy, and corticobasal syndrome [8]. The disease has been considered a subtle and irreversible onset, and every 3 s, a new case of AD is reported [9]. When diagnosed, 67% of the patients have moderate-to-severe disease; the condition progressively progresses at younger ages, posing a serious threat to human health and burdening hundreds of millions of people annually on an economic and social level. According to the latest report, AD is considered a leading cause of death in both rural and urban residents [9]. People with AD frequently have comorbidities like cardiovascular disease because the condition is typically diagnosed in those who are in the mid-sixties or older [10,11].

Although AD has been extensively studied, the precise pathophysiology is still unknown. Numerous theories have been presented to explain the genesis of this disorder, including aberrant Aβ protein accumulation, excessive tau protein phosphorylation, cholinergic neuron dysfunction, oxidative stress, inflammatory reactions, and genetic abnormalities [12]. The diagnostic methods are biological biomarkers [13,14] and imaging analysis.

Given the assumption that AD develops as a series of pathogenic events, a biomarker’s abnormality will have a higher immediate prognostic value for a process that comes just after it in the cascade. Numerous biomarkers have been linked to a higher chance of pathological and clinical development in AD [15]. The complex pathophysiology of AD demands a holistic analysis approach, moving beyond single-biomarker detection toward integrated strategies parallel to genomics, proteomics, and metabolomics. However, high-performance plasma biomarker panels remain costly and technically complex, limiting their use for widespread screening and first-line diagnostics [16]. Tau is a brain-specific multifunctional microtubule-associated protein that is a more useful and advantageous biomarker than amyloid beta, which is an amyloid precursor protein [17]. The most toxic species in AD is thought to be oligomeric Aβ, which could be a target for AD treatment. At the moment, there are no specific cures or therapies for AD. Due to limited treatment options, an accurate and early diagnosis of AD is essential for timely intervention and slowing disease progression. Current AD diagnosis primarily uses PET Aβ-Positron emission tomography and MRI magnetic resonance imaging. Though effective, these methods are costly, not widely accessible, and lack sensitivity in early disease stages. Moreover, traditional diagnostics rely on clinical and neuropsychological assessments that typically detect AD at the last stages after substantial neuronal damage has occurred. Hence, its early detection and treatment are currently crucial strategies for successful postponement [18,19,20].

An emerging and innovative strategy in diagnostics methodologies involves the integration of nanoparticles-based biosensing platforms. The attractive properties of nanoparticles like various shapes and sizes, high surface-to-volume ratio, and the modifying of their surface with variety of ligands have forced them to play their role in the diagnosis of challenging diseases including AD. Moreover, nanoparticles offer a promising approach to overcome several limitations of invasiveness and high costs which are associated with the other used conventional techniques by providing a cheap and advanced alternative tool for the diagnosis of AD. Owing to their ease of integration, nanoparticles can be embedded into miniaturized diagnostic platform as nano-biomaterials, enabling the development of point-of-care preventive diagnostic tool for AD [21,22].

### 1.1. Overview of AD Biomarkers

The clinical studies revealed that variations in the concentration of amyloid-β-(Aβ), tau, and p-tau in the brain’s blood and cerebrospinal fluid (CS fluid) have significant effects during the development of AD. Significant progress has been made in the accuracy, sensitivity, and specificity of blood-based biomarkers for AD [23].

#### 1.1.1. Core AD Biomarkers (AD Neuropathologic Change-ADNCP)

These biomarkers directly reflect hallmark pathological changes in AD:Amyloid-β (Aβ):

As a result, Aβ is now universally acknowledged as a powerful biomarker and target for AD diagnosis and treatment [24]. Variants such as Aβ_40_ and the Aβ_42_/Aβ_40_ ratio are frequently measured using advanced techniques like a single-molecule array (Simoa) [25]. However, Aβ detection is challenging owing to its aggregation tendency and instability, particularly in plasma [26,27]. Moreover, there is a commercially accessible blood kit (Aβ_1-42_/Aβ_1-40_) in the United States; however, it just calculates the risk of developing AD without offering a diagnosis [28].

Tau protein:

A microtubules-associated protein responsible to regulate the movement of vesicles and organelles and promote axon development [29]. In AD, hyperphosphorylated tau forms neurofibrillary tangles, impairing mitochondrial activity and causing synaptic dysfunction [30]. Six tsu isoforms exist (352–441 amino acids, 50–65 KDA) [29].

Phosphorylated tau (p-tau):

Especially, p-tau 181 and 217 are considered the most reliable early diagnostics biomarkers [31]. In healthy individuals, p-tau 181 and 217 are typically below 2 pg/mL and 0.2 pg/mL [32]. In the case of the disorder, the normal levels of p-tau 217 showed a 6-fold increase from its normal level, while p-tau 181 showed an increase of 0.01 pg/mL in its normal level, which are frequently noticeable before cognitive symptoms appear [33].

#### 1.1.2. Emerging and Supporting Biomarkers

These involve proteins and molecules indirectly involved in AD pathology or useful in differentiating AD from other neurodegenerative diseases.

Neurofilament Light Chain (NFL):

A general marker of neuronal damage, often elevated neurodegenerative disease including AD [34].

Metal ions:

Dysregulation of metal ions contributes to oxidative stress and Aβ aggregation [34].

Apolipoprotein E (ApoE):

This plays a crucial role in transport of cholesterol in the CNS. The ε4 allele is strongly linked to increase AD risk, while ε2 is considered a protective and ε3 is neutral [35,36].

#### 1.1.3. Non-Specific or Non-AD Co-Pathology Biomarkers

These are markers that, while not specific to AD, play a role in its progression or Co-pathology.

Lactoferrin (LF):

LF is a multifunctional iron-binding glycoprotein involved in innate immunity, exhibiting anti-inflammatory, antioxidant, and antimicrobial properties. Moreover, it plays a critical role in host defense by inhibiting the growth of pathogens and modulating immune responses. Thus, it is considered an important protein and has a key role in maintaining the human body [37].

Cortisol:

Cortisol is an essential hormone which is discharged by the adrenal cortex. When AD patients are in the early stages or even the preclinical phase of the disease, elevated cortisol levels are linked to a worse prognosis and the rapid progression of cognitive impairment. Furthermore, cortisol may increase oxidative stress and tau and Aβ pathologies, which could contribute to the pathogenesis of AD [38].

#### 1.1.4. Diagnostics Criteria and Techniques

The current biological diagnosis of AD is based on the A/T/N classification:

A: Amyloid deposition (e.g., Aβ_1-42_, Aβ_1-42_/Aβ_1-40_ ratio);

T: Tau pathology (e.g., p-tau 181, p-tau 217);

N: Neurodegeneration (e.g., t-tau, NFL).

Moreover, diagnostic thresholds for these biomarkers in CSF have been established and continue to aid in clinical assessments [28].

Hence, from the above discussion it may be concluded that, core markers like Aβ and tau remain central to diagnosis; supportive and non-specific biomarkers are gaining recognition for their role in enhancing early detection accuracy and understanding AD pathophysiology. However, challenges persist in detecting biomarkers like Aβ_40_ owing to their low plasma concentrations [39].

There is still a special need to inform individuals about the early symptoms, spreading rate, and treatment steps regarding AD. To cover the above scientific gap, in this review, responsible biomarkers, applied nanoparticles and their size, and analytical methods (electrochemical, colorimetric, lateral flow assay, fluorescence, surface-enhanced Raman scattering sensors) along with their advantages and disadvantages, author geographical locations, diagnostic method, and LOD, are studied to investigate the AD-affected population and to pave the path to keep control over it. Numerous biomarkers have been linked to a higher chance of pathological and clinical development in AD [15]. Moreover, the major purpose of this review is to present a comprehensive overview of how nanomaterials are reshaping the diagnostic landscape for AD. We aim to bridge the knowledge gap between the population level trends of AD, the recent breakthroughs in biomarker discovery, and the cutting-edge role of engineered nanomaterials in diagnostic applications. By integrating these dimensions, this review highlights the potential of nanotechnology-driven approaches in addressing the unmet clinical need for early and accessible AD diagnosis. The concept is to provide a multidisciplinary perspective that informs researchers and clinicians about emerging nanomaterials-based diagnostics strategies and their implications for public health.

### 1.2. Biomarker Related Diagnostic Techniques

The use of non-invasive imaging biomarkers for the monitoring and development assessment of AD grips important clinical application. Neuroimaging techniques have the advantages to provide noninvasive insights into brain functionality, metabolic process and anatomy. Several non-invasive and highly sensitive imaging techniques have been explored for the early diagnosis of AD, such as PET, FDG (fluorodeoxyglucose)-PET, magnetic resonance imaging (MRI), and Computed tomography (CT). These methods are valued for their simplicity, rapid response, and accuracy [40,41,42,43,44]. Moreover, it is also considered a novel route to use in the case of high-risk people affected by AD. Prof. E.D. Smith Dphil et al. successfully checked the performance of CT in several AD patients [45]. The concentration of amyloid-β-(Aβ), tau, and p-tau in the brain’s blood and cerebrospinal fluid (CS fluid) have significant effects during the development of AD.

A comprehensive classification of biomarkers and sources has been generated for AD diagnostic relevance in Figure 1.

Firstly, the core biomarkers, highlight two main pathological features: Aβ proteinopathy (core 1) and tau pathology (core 2), further divided into secreted and phosphorylated forms (Figure 1). The next category focuses on CSF or plasma analytes, detailing specific Aβ and tau isoforms, including p-tau 217, p-tau 181, and p-tau 231 in core 1, and additionally phosphorylated and non-phosphorylated tau species in core 2. Moreover, imaging biomarkers are shown, emphasizing PET imaging for Aβ and tau as part of the core classification [46]. The figure continues with biomarkers of non-specific processes involved in AD pathophysiology, like neuronal injury and inflammation, which are measurable through neurofilament light (NFL), GFAP (glial fibrillary acidic protein), and neuroimaging techniques such as MRI or FDG PET. Lastly, biomarkers of non-AD co-pathology are presented including vascular brain injury markers (e.g., infractions detected through CT or MRI) and α-synuclein levels (measured in CSF or plasma), which help in differentiating AD from other neurodegenerative conditions [46]. Overall, the figure offered a structured overview of the multidimensional biomarker framework essential for accurate AD diagnosis and staging. Additionally, it can be seen that the concentration of amyloid-β-(Aβ), tau, and p-tau in the brain’s blood and cerebrospinal fluid (CS fluid) have significant effects during the development of AD, involving core biomarkers of AD neuropathology change (ADNCP), non-specific biomarkers which are responsible for other brain diseases but also involved in causing AD, and common non-AD co-pathology biomarkers [46].

### 1.3. Advanced AD Diagnostic Methods

Since AD-modifying treatments have recently been developed, its early and precise diagnosis has also become more and more important. The most advanced diagnostic methods for AD currently available are time-consuming (neuropsychological evaluation), costly (neuroimaging), and intrusive (cerebrospinal fluid study). Therefore, there is a growing demand for more noninvasive and affordable methods that enable the identification of individuals in the preclinical or early clinical phases of AD, who may be qualified for additional cognitive testing and dementia diagnosis. The development of sensitive technologies that can reliably measure brain-derived chemicals that are present in blood in extremely low concentrations has made this physically feasible [47]. However, Cerebrospinal fluid (CSF) biomarker testing, neuroimaging, and scales are currently the primary techniques for diagnosing AD [48]. Due to its many benefits, including low production costs, simplicity of downsizing, high integration capabilities, and a wide range of applications, several printed sensors like laser-printed microfluidic chip sensor have been used in the ultra-sensitive detection of AD blood biomarkers [20].

Label-free optical biosensors, particularly fiber-optic SPR biosensors, which are used for the detection of AD biomarkers in blood plasma have attracted more attention from researchers because of their exceptional capabilities, which include direct and real-time biological target detection with high sensitivity and specificity, small size, and low cost [49]. Hui Liu. et al. employed an electrochemical-based biosensor by using polyamide/polyaniline carbon nanotubes (PA/PANI-CNTs) for the detection of AD biomarker (Aβ_42_) with the lower detection limit of 30 fg/mL [50]. Electrochemical biosensors have the advantages of sensitivity, economical, speed, and specificity [51]. However, when proteins are present, electrochemical methods may encounter issues such as electrode fouling, which impedes electron transport and creates interference when determining the amounts of free drugs (pharmacologically active) in actual samples [52,53]. Xue Zhang et al. proposed a surface-enhanced Raman scattering which has the advantages of prominent sensitivity and selectivity based on the bases of GNPs for the simultaneous detection of AD biomarkers (Aβ 1-42 and tau proteins) with the detection limit of 3.7 × 10^−2^ nm [54]. Although SERS (surface-enhanced Raman Spectroscopy) is a very surface-sensitive technology, it is material-limited and requires Surface Plasmon Resonances (SPRs) to enhance the Raman signals. Moreover, the SERS techniques faced the problem of stability and reproducibility optimization during the real sample analyses [55]. Another promising technique on the bases of fluorescence imaging techniques was developed by Hengde Li et al. by using *N*,*N*-diethylaniline for the detection of AD biomarkers (Aβ oligomers) with a strong binding affinity (K_d_ = 6.16 nm) [56]. The advantages of fluorescence imaging techniques are their sensitivity, selectivity, technical simplicity, and fast response [57]. The major limitation that occurred in the fluorescence-based detection is the uncertainties in the calibration of the responses [58] which suppress its applications. Additionally, thioflavin T (4-(3,6-dimethyl-1,3-benzothiazol-3-ium-2-yl)-*N*,*N*-dimethylaniline, ThT), is a well-known water soluble fluorescent probe that has long been recognized as a highly sensitive dye for detecting of amyloid structures; when bound to amyloid fibrils, it exhibits fluorescence with specific excitation and emission spectra. Therefore, it is commonly employed to detect protein aggregations related to disease. A ThT-based approach offers significant advantages owing to its simplicity, high sensitivity, and reliable accuracy. However, the major disadvantage of ThT may lead to misinterpretation of experimental results due to electrostatic repulsion. Ding et al. incorporated the fluorescent dye ThT into the three-dimensional metal–organic framework Er-MOF to develop a radiometric fluorescence sensor for the detection of AD biomarkers [59,60,61]. Yanli Zhou et al. developed a colorimetric sensor on the bases of GNPs for the detection of amyloid-β peptide (Aβ) AD biomarkers by achieving the detection limit of 0.6 nM in human blood serum [62]. The advantages of such sensor include its simplicity, low cost, rapidity, and straightforward reactions by observing the sensing just by color change [63,64]. Unfortunately, the colorimetric sensor has the limitation of interference from the background matrix which may be affected by the presence of other colored substances in the sample matrix [65]. Nanoparticles in the colorimetric techniques can aggregate with respect to time, which affects their optical properties and reduces their sensitivity and accuracy. Surface Plasmon Resonance (SPR) has been extensively used for the detection of AD biomarkers like β-amyloid and tau proteins. The advantage of the proposed sensor is to sense the targeted analyte in a very small quantity in the cerebrospinal fluids and blood [66]. The major limitations include the immobilization of biomolecules over the SPR chip surface without disrupting their activity.

Among the aforementioned techniques, lateral flow assays (LFAs) have attracted more attention from researchers towards the detection of disease biomarkers owing to their low cost, quick response, and friendliness behavior. The ASSURED criteria (affordable, sensitive, specific, user-friendly, rapid/robust, equipment-free, and deliverable for end users) are met by LFA in accordance with World Health Organization (WHO) recommendations [67]. Since it does not require any specialized knowledge or training to operate, the LFA sensing method is regarded as a practical point-of-care (POC) diagnostic tool [68]. Sandwich-type and competitive testing procedures are the two primary detection strategies often used in the development of LFAs [69].

Among paper-based POCT (point of care testing), LFAs are one of the best and most widely used. LFA technology has a wide range of applications in several fields such as health, environment, agriculture, and medical-related fields. It is a powerful tool for the monitoring of dengue, cardiac and cancer biomarkers, severe acute respiratory syndrome coronavirus 2, and carcinoembryonic antigen [69]. Following the discovery of human chorionic gonadotropin in the urine of pregnant women, the human pregnancy test popularized the LFA principle as shown in Figure 2.

Xiangwei Zhao et al. projected a SERS-LFA sensor for the simultaneous detection of Aβ 42, Aβ 40, tau proteins, and neurofilament light chain biomarkers by using gold core silica shell (Au@SiO_2_) nanocomposites to achieve the LOD of 138.1, 191.2, 257.1, and 309.1 fg/mL [71]. Lais C. Brazaca proposed a paper-based lateral flow immunoassay by using GNPs@antibodies for the detection of fetuin B and clusterin biomarkers with the LOD of 0.24 nM and 0.12 nM, respectively [72]. Li Wang et al. applied the LFA-based sensor based on GNPs for the detection of miRNA-155 with the LOD of 100 pM [73]. From the above theoretical discussion, it has been concluded that LFA-based sensor techniques have a wide potential towards the detection of AD biomarkers in terms of their achieved LOD values and the researcher’s interests.

By measuring the color shift brought on by the buildup of GNPs, the LFA results are typically visible to the naked eye. Although this detection method is quick and easy, the results are qualitatively insensitive, and it might only work in specific situations. However, these approaches do not provide adequate sensitivity for the detection of essential biochemical markers that are present in very minute concentrations in a sample, which limits their uses [74]. Another drawback is that the results can become subjectively interpreted visually, especially if the end user is relied upon to interpret several test lines or detect different line intensity gradients [75]. Hence, one of the frequent problems mentioned in the usability studies for LFA testing seems to be the challenges in the interpreting of data [76].

## 2. Research Method

### 2.1. Search Procedures

This systematic review was conducted following the PRISMA 2020 guidelines. A comprehensive search of Web of Science, ScienceDirect, and Google Scholar was performed in May 2025 to identify peer-reviewed literature published between January 2021 and May 2025.

Search strings included combinations of the following keywords using Boolean operators “AND” and “OR”:(“Alzheimer’s disease” OR “AD” OR “dementia”) AND (“biomarkers” OR “diagnostic method” OR “LOD” OR “limit of detection”) AND (“nanoparticles” OR “techniques”). The search focused on studies reporting limits of detection (LOD) for biomarkers such as ApoE, Aβ_40_, Aβ_1-42_, iRNA-16, dopamine, tau proteins, p-tau proteins, AβO biomarkers, serotonin, and acetylcholinesterase (AChE) and various employed analytical techniques including electrochemical, colorimetric, lateral flow assay, fluorescence, and surface-enhanced Raman scattering sensors. Filters were applied to restrict results to English-language journal articles with full texts available.

### 2.2. Study Selection and Quality Assessment

Two independent reviewers screened titles and abstracts to remove irrelevant studies. Full text articles were then evaluated based on the inclusion and exclusion criteria. The quality of included studies was assessed using PRISMA 2020 checklist for analytical studies, focusing on clarity of methodologies, reproducibility of results, and robustness of reported detection limits. Disagreements between reviewers were resolved by consultation with a third reviewer.

### 2.3. Risk of Bias

Potential sources of bias were addressed by

Including only peer-reviewed articles to reduce publication bias;Applying strictly the inclusion/exclusion criteria to minimize selection bias;Conducting independent duplicate screening and quality assessment to reduce reviewer bias.

### 2.4. Inclusion and Exclusion Criteria

Inclusion criteria

Studies published between January 2021 and September 2025 to ensure the inclusion of the most recent advances.Articles reporting original research on diagnostic biomarkers or techniques for AD.Provided quantitative outcomes, particularly specifying the limits of detection (LOD) for at least one AD biomarkers.Articles published in peer-reviewed journals and indexed in at least one major database (e.g., PubMed, Web of Science, Scopus).Available in full-text format in English language.

Articles were excluded if they were

Published in languages other than English;Limited to abstracts only, without access to full text;Non-original research articles such as meta-analyses, conference abstract, or editorials, etc.;Studies focusing on diseases other than AD or that did not evaluate diagnostics biomarkers relevant to AD;Articles that did not provide measurable analytical performance (e.g., LOD, sensitivity, or specificity).

### 2.5. Statistical Analysis

To compare the analytical performance of sensors reported during two periods (2012–2020, *n* = 13; 2021–2025, *n* = 20), the lowest 13 LOD values from each group were analyzed. Since the data were not normally distributed, the Wilcoxon unpaired test was employed to assess difference in LODs between the two timeframes. For consistency, duplicate or misclassified values were corrected before analysis. After which, the average LOD for each sensor type was calculated across both timeframes and confirmed that the 2021–2025 period consistently demonstrated lower LODs compared to 2012–2020.

### 2.6. Research Questions

The review was designed to address the following key research questions:What is the global distribution of AD across populations by age and geographic regions?What emerging biomarkers are being explored for the diagnosis of AD?What advancements in diagnostic techniques for AD have been developed recently?

These questions guided the systematic review and formed the framework for data extraction and analysis.

The software tools used for this review were integrated to enhance accuracy and visualization. Microsoft 365 was employed for graphing and summarizing data, while EndNote X9 (version 2021) was used for managing and formatting references. Figures, including the PRISMA flow diagram, were designed using PowerPoint (version 2016).

## 3. Results and Discussion

### 3.1. Screening Process

Articles were screened in adherence to the Preferred Reporting Items for Systematic Reviews and Meta-Analyses (PRISMA) guidelines in Figure 3.

The screening followed these steps:Initial Filtering: Titles and abstracts were reviewed to remove irrelevant articles.Categorization: Articles were classified into:Primary Articles: Reporting original experimental or observational data.Methods Papers: Describing or evaluating techniques for biomarker detection.

### 3.2. Data Extraction and Reporting

A standardized data extraction form was used to collect information on study location, diagnostic methods, limits of detection, nanoparticles applied, and biomarkers analyzed. The review included 33 studies, categorized into three groups as shown in Figure 4.

The iridium oxide nanoparticles (IrO_2_) have attracted much attention form the researchers in the area of electrochemical sensor owing to their excellent electrocatalytic activity. Moreover, the novel IrO_2_-NPs-based electrocatalytic assay enables signal generation directly in the immunoassay medium, eliminating the need of extra reagents and supporting future integration into biosensing platforms for proteins, DNA, and cell analysis [77]. Moreover, DNA nanotechnology serves as a versatile synthetic platform for engineering customized DNA nanostructures (DNS), offering extensive potential for applications across diverse scientific and technological domains [78]. The reasons behind the selection of gold nanoparticles (GNPs) in the references [79,80] are their excellent conductivity and catalytic behaviors which increased the sensitivity of the electrode by three orders of magnitude. Furthermore, GNPs offer outstanding stability, a large surface-to-volume ration, and strong resistance to matrix-induced degradation. The cadmium-selenide/zinc-sulfide (CdSe@Zn) has several unique properties and is used for the purpose of biolabeling capability and for the improving of sensitivity [81]. The carbon (nanodiamond) is used in the reference [82] owing to sensitive behaviors which have attracted much attention of the researchers in the area of diagnostics of challenging illness. The MoS_2_ nanoparticles were employed in the reference [83] because of the quick and strong adsorption of single-strand DNA (ss-DNA) over the surface of MoS_2_ nanoparticles to enhance the fluorescence signal. Behind conventional nanoparticles, DNA-based nanostructures are increasingly attracting significant interest within the biomedical field [84]. The manganese dioxide (MnO_2_) nanoparticles used in the reference [85] showed the outstanding peroxidase-mimicking activity owing to their high surface area. In the case of reference [77], the obtained LOD of 68 ng/mL was considered high which may be due to the lower surface area and limited available binding sites for the interaction of the investigated biomarkers. The GNPs in the case of the detection of dopamine showed a high LOD of 50 ng/mL by using the LFA sensor technique due to the interfering of the other biomarkers present in the urine samples. In the same way, the high LOD of 67.71 ng/mL and 56.2 ng/mL, which was obtained by the Ag and polymer nanoparticles, towards the detection of Aβ_42_ and AβO by using the SERS and fluorescence techniques, may be due to any possible technical reasons.

Table 1 provides a detailed literature review by giving information from a total of 33 published research articles regarding detected biomarkers of AD, applied nanoparticles, obtained LOD (ng/mL), the country of the authors, and the employed analytical techniques. E. Sensor, LFA, F. Sensor, SERs, and C. Sensor have been explored to detect Alzheimer’s disease. The techniques, biomarkers, applied nanoparticles, LOD (ng/mL), and country of authors are listed in Table 1. Moreover, the values for particle size (nm) presented in Table 1 correspond to the hydrodynamic diameter obtained from dynamic light scattering (DLS), which includes the solvation layer surrounding the particles in suspension. The value is typically larger than the core dimensions observed by electron microscopy technique which reflects the actual molecular or crystalline size. Hence, the size reported should be interpreted as the effective hydrodynamic dimensions relevant under aqueous conditions.

### 3.3. Comparison of Nanoparticles and Biomarkers

Figure 5a,b, which is plotted from the data in Table 1, is shown to compare the applied nanoparticles and the investigated AD-related biomarkers. According to the concept of Figure 5a, 16 (48.5%) research articles are concerned with GNPS, followed by Ag 4 (12.1%), Polymer 2 (6.07%), and Magnetic 2 (6.06%), while 27.27% cumulatively account for pyrrole 1 (3.03%), CH-Cu-N-As 1 (3.03%), CdSe@ZnS 1 (3.03%), EuUPDC 1 (3.03%), IrO_2_ 1 (3.03%), carbon 1 (3.03%), LDMD-N 1 (3.03%), MoS_2_ 1 (3.03%), and MnO_2_ 1 (3.03%) nanoparticles. Figure 5b showed that the most studied biomarkers are Aβ_42_ which has 7 research articles (21.2%), Aβ_1-42_ which has 4 (12.1%), AβO which has 4 (12.1%), p-tau proteins which have 4 (12.1%), Aβ_40_ which has 3 (9.09%), Dopamine which has 3 (9.09%), Aβ_40_ which has 3 (9.17%), and ApoE which has 2 (6.06%); while cumulatively of 9.09%, miRNA-16 has 1, Aβ_1-40_ has 1, and AChE has 1, respectively. From the above discussion, it is concluded that the most active nanoparticles used for the monitoring of AD are GNPs owing to their unique properties like exceptional optical properties, nano-scaled chirality, nano-amyloidosis, and antioxidant effect [112], while in the case of biomarkers, these are Aβ_40_, Aβ_42_, dopamine, p-tau proteins, and AβO biomarkers, due to their direct linkage to the pathological process, considered as potential targets for the early diagnosis.

As shown in Figure 6a, the most research articles regarding the detection of AD biomarkers are published in the years of 2025, followed by 2024, 2023, 2022, and 2017. Moreover, some number of articles in favor of detection of AD biomarkers are also published in the years of 2012, 2013, 2014, etc. Hence, it may be argued that in the present decade, researchers are interested in monitoring AD on time.

Figure 6b gives information about the countries of researchers with respect to number of articles. The highest number of articles (19) are published by the scientists of China, followed by Spain, USA, and South Korea. Hence, it may be concluded that the researchers of China are most active in diagnosing AD biomarkers.

Figure 6c gives information about the applied bio-recognition elements with respect to published articles. It is detected that mostly used bio-recognition elements are antibodies, followed by neurotransmitters, nucleotides, and other proteins. The reason for the use of antibodies is due to their outstanding performance in achieving the lowest LOD.

Figure 6d presents the trend of LOD achieved by different sensor technologies. The distribution of data points across the years reflects the number of publications and evolution in detection performance. By comparison, it shows that electrochemical sensors appear most frequently across the timeline, indicating a maximum number of publications in the area of AD biomarker detection and continuous research activity. In contrast, the other techniques like LFA, fluorescence, SERS, and colorimetric appeared less frequently, suggesting fewer publications. However, after 2021, there is a noticeable lower LOD in studies involving diverse techniques, highlighting research progress and growing interest in developing highly sensitive diagnostic platforms. Moreover, the Wilcoxon test yielded a statistically significant difference before and after year 2021 (*p* = 0.029). It has confirmed improvements in detection sensitivity in recent years [113].

Table 2 summarizes common AD biomarkers including Aβ_42_, Aβ_42_/Aβ_40_ ratio, t-tau, and p-tau 181 along with their typical concentration ranges in CSF and plasma, as reported in clinical studies [114,115,116,117].

As displayed in Table 2. Aβ_42_ levels in CSF are typically reduced in AD patients with a diagnostic cutoff around <495–500 pg/mL [114]. The ratio of Aβ_42_ to Aβ_40_ is considered more reliable than Aβ_42_ alone, with pathological threshold of ~0.07–0.10 in CSF [115] and ~0.0450–0.0472 in plasma [116], indicating amyloid deposition in the brain. Tau protein serves as a complementary biomarker: t-tau levels in CSF rise significantly in AD, with a cutoff of <266 pg/mL [117], while p-tau 181 is considered a more specific biomarker of tau pathology, which is typically below ~26.59 pg/mL in healthy individuals but elevated in AD [117]. Together, these biomarkers provide a robust diagnostic profile for distinguishing AD from normal aging and other neurodegenerative conditions.

### 3.4. Comparison of the Performance of the Applied Techniques and Nanoparticles with Respect to Years

The data from Table 1 showed the relation of the applied nanoparticles and used techniques with respect to the years. An attractive trend between the used techniques with respect to time (2012–2025) has been achieved. The result indicated that the techniques of LFA, SERS, and colorimetric sensors were mostly used during the period of 2012–2025, while the other electrochemical and fluorescence sensors received less attention from researchers in the same time period. Moreover, Table 1 further gives information about the applied nanoparticles in the period from 2012 to 2025. The trend is extremely competitive by showing the outstanding performance of the magnetic NPs, followed by carbon (nanodiamond), GNPs, AgNPs, and IrO_2_ nanoparticles in the period of 2012–2025. The other nanoparticles, including polymers, CdSe@ZnS, MoS_2_, and MnO_2_, are also significantly used by the researchers from 2012 to 2025 but have not achieved the desired results. Besides the outstanding performance of the current analytical techniques (LFA, colorimetric, electrochemical, fluorescence, and SERS), they are facing several limitations during the detection of biomarkers including low sensitivity, semi-quantitative results, reproducibility, costly labeling, matrix interference in real samples, and restricting of clinical applicability [118,119]. Hence, to cover these challenges, hybrid nanomaterials-based platforms, signal amplification strategies, and integration with microfluidics and AI-assisted data integration are being explored to improve sensitivity, reproducibility, and portability [120]. Moreover, multimodal diagnostics strategies that integrate complementary technologies could improve sensitivity and robustness while maintaining user-friendliness.

### 3.5. Comparison of the Performance of AD Biomarkers and Nanoparticles Size (nm)

Table 1 confirms that the best result was obtained by using the biomarkers of tau proteins, followed by p-tau proteins, Aβ_1-42_, Aβ_40_, serotonin, miRNA-16, dopamine, AβO, Aβ_42_, AChE, and ApoE. Moreover, with respect to various particle sizes (nm), different techniques are used. The best result was achieved by using carbon-containing nanoparticles with size of 120 nm, GNPs with the particle size 37.3 nm and 35 nm, GNPs having the particle size of 5.7 nm, and MnO_2_ nanoparticles with particle size of 220 nm.

### 3.6. Global Burden and Epidemiological Trends of AD

Various countries have shown momentous research interest in the monitoring of AD. Countries like Turkey, Germany, Portugal, South Korea, and Russia are actively engaged. Their contributions reflect a strong commitment to advancing diagnostics technologies for AD. Hence, these efforts highlight the global recognition of AD as a critical health challenge and shared objective of improving early diagnosis through innovative research. In the middle of the 1980s, Alzheimer’s disease was initially identified by neuropathology, which eventually evolved into a more accurate neuropathological diagnostic that recognizes the coexisting neuropathology that typically contributes to clinical dementia and is considered a progressive and irreversible disorder [121]. About every five years, the number of elderly individuals with Alzheimer’s doubles, and by 2060, the frequency is predicted to quadruple [122].

In the present decade, around ten million new cases of neurodegenerative diseases are identified every year, of which 70% suffer from AD [123]. In 2015, millions of people of all ages with AD were diagnosed, which has doubled every 20 years. Major factors such as oxidative stress, abnormalities in cholinergic neurons, and inflammatory cascades mainly contribute to the instigation and propagation of AD. Moreover, abnormalities in the metabolism of biomolecules like proteins, fats, and carbohydrates are considered mainly responsible for numerous metabolic-related disorders, such as metabolic syndromes (MetS), obesity, and type2 diabetes mellitus (T2DM), which are also connected with the insults of the CNS, especially the pathology of AD [124]. An estimated 57 million individuals worldwide suffered from dementia in 2019, which is the most prevalent type, and by 2050, that number is expected to reach 153 million [125]. According to the previous report, it is the fifth most common cause of death. Without showing any specific symptoms, people with AD can be alive for up to 20 years [126].

The National Institute of Aging [127] estimates that AD is the most common type of senile dementia and one of the main causes of death in the world, especially in the US, which affects approximately 6.7 million older Americans [128]. According to the US Alzheimer’s report of 2019, the total number of affected persons by AD is 6.7 million, with a total treatment cost of USD 340 billion. Up to 2050, it has been expected to affect 14 million people, with the enhancement in the treatment costing up to USD 1.1 trillion [129]. The United States was able to better comprehend the associated economic burden and assist legislative efforts to more effectively treat ADRDs after Hurd et al. published a critical study quantifying the cost of dementia sickness in the country in 2013 [130]. An Australian report delineated that approximately 472,000 citizens live with AD with a total treatment cost of $1.98 billion (2018–2019), which is expected to increase to 598,000 individuals in 2028 [131]. Furthermore, in China, among people 60 years of age and older, the prevalence of dementia and moderate cognitive impairment (MCI) is 6.0% and 15.5%, respectively. The majority of those cases (65%) are dementia or AD-related MCI, with vascular dementia (26.7%) and other dementias (8.3%). In China, almost 260 million people are 60 years of age or older. In 2019, AD was considered the fifth most common cause of death in China. By 2050, China will have spent USD 1.89 trillion on nursing care and dementia treatment [132].

## 4. Challenges and Future Perspectives

The on-time monitoring of AD with the help of diagnostic tools on the basis of effective nanoparticles have attracted more attention from researchers. Plasma p-tau231 and p-tau217 are blood-based biomarkers strongly linked to AD pathology, reflecting both early Aβ buildup and advanced tau deposition [133]. Although blood-based biomarkers testing is less invasive, more accessible, and cost-effective than PET imaging or lumbar puncture [134], its clinical application is hindered by the ultra-low concentrations of biomarkers, and the difficulty is differentiating pathological signals from normal physiological levels [135]. Moreover, identifying accurate and specific biomarkers from neuroimaging data presents a significant challenge, particularly when aiming to characterize distinct populations and brain regions [136]. Likewise, obesity and metabolic syndrome have been linked to reduced plasma NFL concentrations, potentially attributable to increased blood volume in affected persons, which may dilute biomarkers levels. However, t-tau concentrations show a positive correlation with body index, indicating that metabolic factors may influence specific biomarkers in distinct ways [137,138]. Moreover, to provide a platform for more precise and trustworthy biomarker developments, the wealth of data produced by non-invasive methods like blood component monitoring, wearable sensors, imaging, and biosensors also dramatically lower patients’ pain, risk of complications, psychological impact, and expenses. However, there is the challenge of in-computationally analyzing the vast amount of data produced which can yield vital information for early diagnosis of AD [139]. Additionally, biosensors and portable devices often face challenges with reproducibility and long-term stability, particularly when used in diverse real-world environments outside controlled laboratory settings [140].

Future perspectives: Developing techniques in nanotechnology and artificial intelligence are gifts to increase sensitivity, allowing for the detection of AD biomarkers in simple biofluids like sweat, blood, and saliva before symptoms appear. To better understand health inequities, researchers may now examine huge sets of highly correlated biomarkers in addition to gathering and compiling their data with the help of computational achievements like machine learning techniques. Moreover, to understand educational health inequities in disability, morbidity, and mortality, for instance, researchers have begun integrating epigenetic clocks, polygenic risk scores, and other multifaceted measures of aging [141,142,143]. Future research may be influenced by the promising prospect presented by the combination of machine learning with blood spectral test technology for creating cutting-edge illness predication techniques and enhance patient outcomes by using more accurate predication techniques. As machine learning-based models have been developed, maintaining interpretability in diagnostics decision-making continues to be a top priority [16]. These improvements could enable timely involvements that control disease evolution, improving outcomes for patients, and reducing healthcare costs. A diagnostic tool such as LFA that becomes more accessible and precise holds the potential to transform Alzheimer’s care and guide the route toward the future to detect neurodegenerative diseases with unprecedented precision.

Additionally, early identification of AD through retinal biomarkers enables targeted interventions, including personalized lifestyle strategies like neuroprotective dietary adjustments, structured physical activity, and cognitive stimulation which are proven to reduce risk and decelerate disease progression. Early detection of AD biomarkers could critically enhance the timely administration of emerging disease-modifying therapies targeting Aβ and tau pathology [144]. Future research should prioritize the standardization of blood-based biomarker assays, the definition of universally accepted diagnostic thresholds, and the implementation of large-scale, multi-population studies to confirm their diagnostic and prognostic accuracy. Combining blood-based biomarkers with advanced neuroimaging and Al-driven analytical tools can significantly improve diagnostic accuracy, enabling more precise disease staging and longitudinal monitoring. The use of mobile and wearable technologies seems appropriate for the monitoring of neurological disorders including AD. The potential of home polysomnographic monitoring makes sense to look for early symptoms in order to treat these conditions before their clinical symptoms appear [145]. The objectives of mobile health is the integration and centralized administration of health information. The main advantages of employing mobile health include accessibility, self-management, real-time monitoring, and cost-effectiveness. Wearable technology is considered a vital source of data for this process. These gadgets gather health information that is directly relevant to the user’s everyday activities and work with mobile health services and apps to provide real-time monitoring and individualized treatments [146]. Many portable biosensing platforms promise affordability, large scale productions, reagent stability, and quality assurance, which remain critical barriers to maintaining cost-effectiveness without compromising sensitivity. Portable devices should not be seen as replacements but as complementary tools that integrate with established clinical workflows and ensure interoperability and validation against gold-standard methods, which is crucial for clinical acceptance. Moreover, the ability of blood-based biomarkers to detect asymptomatic individuals at risk for AD offers promising prospects for early therapeutic intervention, potentially delaying disease progression and mitigating the global impact of AD [147]. Moreover, biomolecule-induced liquid interfaces at micro/nanoscale enable advanced insights and innovations in visual biosensing. This liquid interface-based biosensing technology enables rapid, low-cost, and self-administered disease detection and monitoring, making it more suitable for resource-limited healthcare settings [148]. To further cover the barriers towards the detection of AD biomarkers, along with solid, liquid nanomaterials have also attracted much attention from researchers owing to several unique properties like antifouling, self-healing, optical, and adaptive behavior [149]. Additionally, liquid crystals have the potential to be highly used for biosensing purposes because of their anisotropic nature and quick responsive behaviors [150].

## 5. Conclusions

This review provides detailed information about the AD population or distribution, performance of various nanoparticles (magnetic, GNPs, carbon, Ag, IrO_2_, polymers, MoS_2_, MnO_2_, etc.), biomarkers (Aβ_40_, Aβ_1-42_, Aβ_42_, Serotonin, AChE, dopamine, p-tau proteins, AβO, etc.), analytical techniques (electrochemical sensor, colorimetric sensor, LFA, SERS, fluorescence, etc.), obtained results, and country of authors from 2012 to 2025. By considering the results, the desired result towards AD detection was usually showed by GNPs followed by, silver (Ag) polymers, and magnetic and other materials, by using Aβ_40_, Aβ_1-42_, Aβ_42_, dopamine, p-tau proteins, and AβO biomarkers, respectively, with the help of LFA, SERS, and colorimetric sensor techniques. In terms of particle size, the best result was obtained with the help of GNPs, silver, and magnetic and carbon-containing nanoparticles. To further improve the worth of the current work, the methodology used for the collection of the data has been discussed in the form of PRISMA diagram. Moreover, the interest of the various participating countries including China, Germany, Taiwan, South Korea, Portugal USA, Spain, Russia, Turkey, etc., has also been included. The result obtained from the plotted information favored the performance of Portugal, Germany, and Turkey. The comparison of LOD with respect to applied analytical techniques was investigated and the electrochemical sensor was highly recommended. Hence, it may be concluded that this review also has the potential to make collaboration easier for researchers around the world to achieve the desired results and to cover the challenge of AD diagnosis in clinical bases.

## Figures and Tables

**Figure 1 ijms-26-09282-f001:**
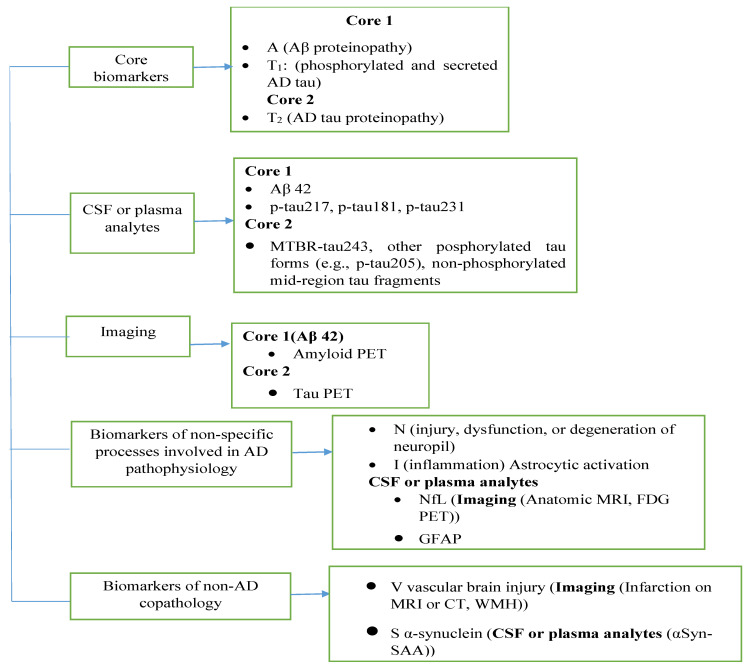
Classification of fluid analyte and imaging biomarkers. Abbreviation: α-Syn-SAA; alpha-synuclein seed amplification assay, Aβ; amyloid beta, NFL; neurofilament light chain, GFAP; glial fibrillary acidic protein, MTBR; microtubule-binding region.

**Figure 2 ijms-26-09282-f002:**
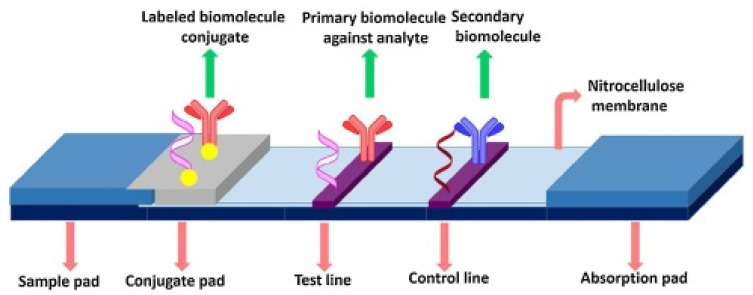
Basic principle of LFA sensor, reproduced/reprinted with permission from reference [70] (Copyright-License number 6090601339315).

**Figure 3 ijms-26-09282-f003:**
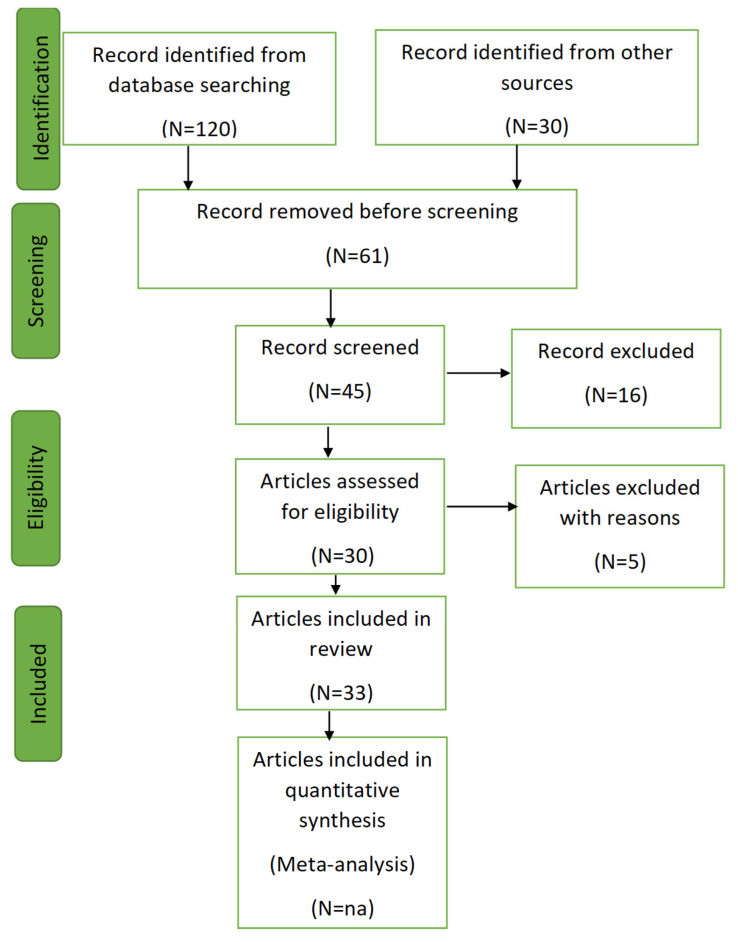
PRISMA flow diagram for literature review; na = not applicable.

**Figure 4 ijms-26-09282-f004:**
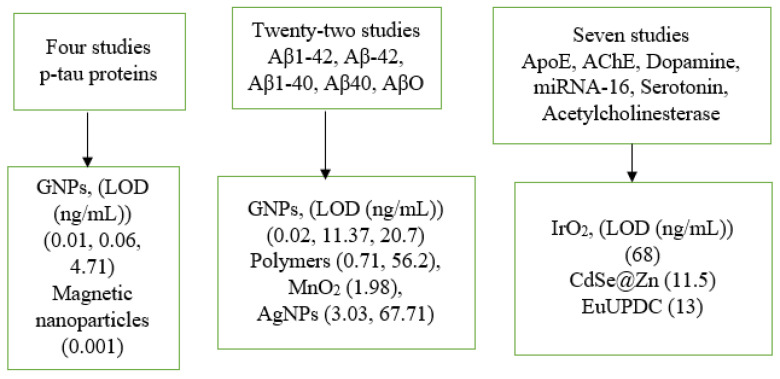
Nanoparticles, biomarkers, and LOD-based categories of 33 review articles.

**Figure 5 ijms-26-09282-f005:**
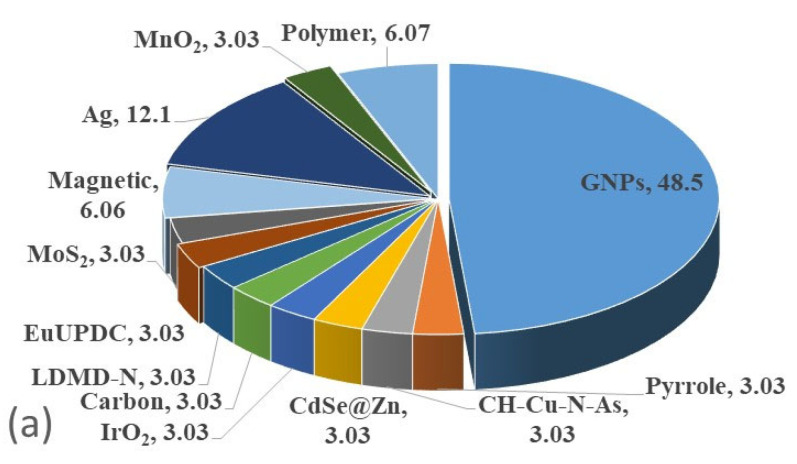
Percentage of studies involving the detection of Alzheimer’s disease by different types of (**a**) particles and (**b**) biomarkers.

**Figure 6 ijms-26-09282-f006:**
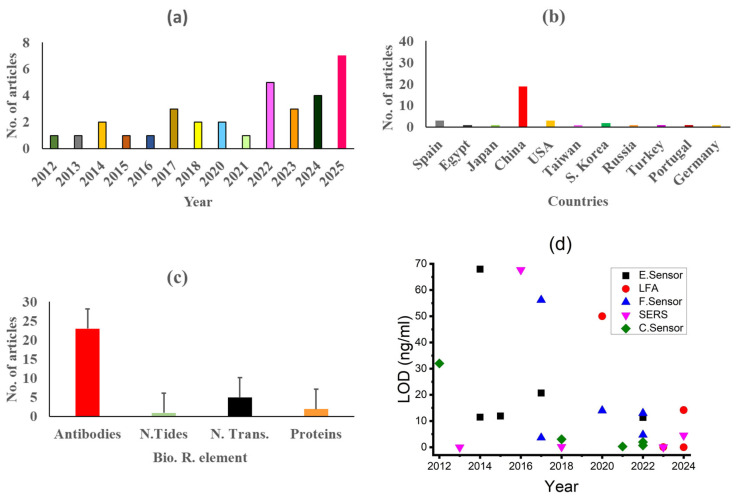
(**a**) Comparison of years with respect to published articles, (**b**) comparison of countries with respect to published articles, (**c**) comparison of bio-recognition elements with respect to published articles, and (**d**) timeline of LODs on the detection of Alzheimer’s disease by different techniques.

**Table 1 ijms-26-09282-t001:** Review various sensing techniques with the obtained LOD and applied nanoparticles.

Techniques	Biomarkers	Biore- Cognition Element	Particles	Size nm	LOD ng/mL	Country of Authors	Year	Ref.
E. Sensor	Aβ_1-42_	Antibody	GNPs	-	0.0001	China	2025	[79]
	Aβ_42_	Antibody	GNPs	-	0.00836	USA	2025	[80]
	Aβ_42_	Antibody	pyrrole	-	0.04	Spain	2025	[86]
	AβO	Antibody	CH-Cu-NAs	-	0.96	China	2025	[87]
	Aβ_42_	Antibody	GNPspoly	45	11.37	China	2022	[88]
	ApoE	Protein	CdSe@Zn	13.5	11.5	Spain	2014	[81]
	Aβ_1-42_	Antibody	GNPs	45	11.93	Japan	2015	[89]
	Aβ_40_	Antibody	GNPs	-	20.7	Egypt	2017	[90]
	ApoE	Protein	IrO_2_	12.5	68	Spain	2014	[77]
LFA	p-tau proteins	Antibody	Carbon	120	0.007	Taiwan	2024	[82]
	tau proteins	Antibody	GNPs	37.3	0.01	China	2024	[91]
	p-tau proteins	Antibody	GNPs	35	0.06	China	2023	[92]
	miRNA-16	N. Tides	GNPs	15	14.2	China	2024	[93]
	Dopamine	N. Transm.	GNPs	-	50	USA	2020	[94]
F. Sensor	p-tau 181	Antibody	GNPs	-	0.00082	China	2025	[95]
	AβO	Antibody	Magnetic	30	3.6	China	2017	[96]
	p-tau proteins	Antibody	GNPs	4	4.71	South Korea	2022	[97]
	Aβ_42_	Antibody	LDMD-N	-	10.86	China	2025	[98]
	Serotonin	N. Transm.	EuUPDC	-	13	China	2022	[99]
	AβO	Antibody	MoS_2_	200	14	China	2020	[83]
	AβO	Antibody	Polymer	80	56.2	China	2017	[100]
SERS	p-ta protein	Antibody	Magnetic	-	0.001	Turkey	2013	[101]
	Aβ_1-42_	Antibody	GNPs	-	0.02	China	2023	[102]
	Aβ_42_	-	GNPs	-	0.1	China	2025	[103]
	Aβ_1-40_	-	GNPs		0.1	China	2022	[104]
	Dopamine	N. Transm.	Ag		0.15	South Korea	2023	[105]
	Aβ_42_	Antibody	Ag	-	4.5	China	2018	[106]
	Aβ_42_	Antibody	Ag	61	67.71	Russia	2024	[107]
C. Sensor	Dopamine	N. Transm.	GNPs	5.7	0.3	Germany	2016	[108]
	Aβ_40_	Antibody	Polymer	-	0.71	Portugal	2021	[109]
	Aβ_1-42_	Antibody	MnO_2_	220	1.98	China	2022	[85]
	Aβ_40_	Antibody	Ag	15	3.03	USA	2018	[110]
	AChE	N. Transm.	GNPs	13	32	China	2012	[111]

Abbreviations: E. Sensor: Electrochemical sensor, LFA: Lateral flow assay, F. Sensor: Fluorescent sensor, SERS: Surface-enhanced Raman spectroscopy, C. Sensor: Colorimetric sensor, ApoE: Antiapolipoprotein E antibodies, N. Tides: Nucleotides, N. Transm: Neurotransmitter, AChE: Acetylcholinesterase. Size: Particle size (nm).

**Table 2 ijms-26-09282-t002:** Representative clinical threshold (literature).

Biomarkers	Matrix	Threshold (pg/mL)	Ref.
Aβ_42_	CSF	~<495–500	[114]
Aβ_42_/Aβ_40_	CSF	~0.07–0.10	[115]
Aβ_42_/Aβ_40_	Plasma	0.0450–0.0472	[116]
t-tau	CSF	~<266	[117]
p-tau 181	CSF	~<26.56	[117]

## Data Availability

Not applicable.

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
