# Peer review of "Molecular Biomarkers for Early Detection of Alzheimer’s Disease and the Complementary Role of Engineered Nanomaterials: A Systematic Review"

_ijms, 2025, doi:10.3390/ijms26199282_

Round 1

Reviewer 1 Report

Comments and Suggestions for Authors

This review article titled “Molecular Biomarkers for Early Detection of Alzheimer’s disease and the Complementary Role of Engineered Nanomaterials: A Systematic Review” provides a systematic review of research into the early detection of Alzheimer’s Disease (AD), particularly focusing on advances in molecular biomarkers and the auxiliary role of engineered nanomaterials in diagnosis, which hold significant practical relevance and academic merit. The manuscript effectively underscores cutting-edge progress in AD biomarker research and the promising potential of nanomaterials to transform diagnostic practices, while also proposing future directions for developing portable, multi-biomarker diagnostic platforms. However, before considering this manuscript for publication, the authors should consider the following points in any revision as follows:

  1. While the manuscript identifies “AI + nanotechnology + multi-biomarker detection” as an emerging trend, the discussion remains overly general and lacks concrete examples or feasible implementation pathways. The authors should strengthen this section by (i) citing recent studies that combine machine learning with blood-based biomarkers, and (ii) providing more specific discussion on the concept of portable diagnostic devices, for instance, their integration with wearable sensors or home-based testing kits.
  2. Although the authors mention following PRISMA 2020 guidelines, the methodology section lacks sufficient detail. Please specify inclusion/exclusion criteria more clearly, outline how studies were assessed for quality, and describe how bias was addressed.
  3. Figures and tables present useful summaries (e.g., Table 1, Figure 5), but analysis is largely descriptive. A more critical comparison (e.g., LOD ranges, cross-technology comparisons, alignment with clinical thresholds) would strengthen the manuscript.
  4. Double check all the reference format. Ensure uniform formatting of DOIs and citation style.
  5. Make sure all abbreviations are written out in full the first time used. This is particularly important in the abstract and in the conclusions, but work through the entire manuscript carefully from this perspective.
  6. Some statements lack adequate citation support (e.g., lines 58–60, the claim that a new global AD case occurs every three seconds).
  7. Language should be polished. Normative and rigorous expression need to be used in review papers. The authors should carefully revise the whole manuscript to avoid grammatical errors, vague sentences, and incorrect format.
  8. The clarity of the figures should be improved, and the text formatting within them (e.g., Figures 5 and 6) should be standardized to ensure consistency across all figures.
  9. In the future perspectives section, it is recommended to discuss more specifically the challenges and prospects of technology translation, such as the detection stability, cost control, and complementarity with existing methods that portable devices may face in clinical applications.
  10. The manuscript should also discuss the limitations of current technologies (e.g., background interference, long-term stability issues) and propose possible solutions. Moreover, multimodal diagnostic strategies could be explored to promote clinical translation.
  11. The manuscript summarized several nanomaterial-based techniques, but the coverage of advanced biosensing strategies is incomplete. For instance, liquid-based interfacial materials represent a highly emerging direction for visual biosensing, and the authors should consider analyzing the feasibility of applying such strategies to Alzheimer’s disease detection, with reference to works including but not limited to Commun. 2022, 13, 1906; Chem. Mater. 2014, 26, 698708.

Reviewer 2 Report

Comments and Suggestions for Authors

I have read the manuscript and I am a little bit perplexed because it is not a scientific work. The manuscript is a review of scientific articles dedicated to the selection of useful biomarkers for the early diagnosis of Alzheimer Disease, AD, and to the selection of specific nanoparticles suitable for the detection of such AD biomarkers. The organization of the material is not bad, but it needs improvement to be understandable to interested readers. In addition, the English is not good and contains unacceptable errors. For instance, the heading for column 3 in Table 1 is “Biorecogenation element” instead of “Biorecognition element”.

The manuscript should be read by a native English speaker with a scientific background to provide helpful suggestions to the authors.

Comments on the Quality of English Language

The English must be improved.

Reviewer 3 Report

Comments and Suggestions for Authors

The manuscript titled “Molecular Biomarkers for Early Detection of Alzheimer’s disease and the Complementary Role of Engineered Nanomaterials: A Systematic Review” by Ul Haq, M.Z.; et al. is a Review work where the authors outlined the most recent advances in the field of the design and development of high-throughput tools for the sensing of those key biomolecular markers for the onset and progression of Alzheimer’s disease. This is a topic of growing interest and the manuscript is generally well-written.

However, it exists some points that need to be addressed (please, see them below detailed point-by-point) to improve the scientific quality of the submitted manuscript paper before this article will be consider for its publication in the International Journal of Molecular Sciences.

1) Introduction. “Alzheimer’s disease (….) affects millions globally, with a significant societal and economic burden (…) mostly affects adults aged 65 years and older and total of 57 million people globally affected” (lines 35-45). Could the authors provide quantitative data insights according to the mentioned economic impact and also the disability-adjusted life years (DALYs) linked to Alzheimer’s malignancies? This will significantly aid the potential readers to better undersand the significance of this devoted Review work.

2) “1.3. Advanced AD diagnostic methods” (lines 226-321). Here, it should be also mentioned thioflavin T (ThT) fluorescence assays to monitor the growth of amyloid fibrils with the main advantages and drawbacks of this technique compared to the other discussed methods.

3) “integration into biosensiong platforms for proteins, DNA and cell analysis (…) The MoS2 nanoparticles was employed (…) quick and strong adsorption of single strand DNA (…)” (lines 380-393). Here, it may be opportune how self-assembled DNA particles [1] can play a pivotal role in the modulation of amyloid aggregation in vivo [2]. This will strengthen the methodology covered by the authors to sense the biomolecular markers that can trigger the onset of Alzheimer’s diseases. Finally, the authors also need to fix the following statement “MoS2 nanoparticles was (…)” by “MoS2 nanoparticles were (…)”.

[1] https://doi.org/10.1021/jacs.4c12637

[2] https://doi.org/10.3390/nano10112200

4) Table 1 (line 410). Does the particle size rely on the real molecular dimensions or it refers to the hydrodynamic corona syze? Some insights should be furnished in this regard. Then, is it completely neccesary to list the information related to the country of the authors in this Table? Maybe the content of this column could be exchanged for other more relevant information like the selectivity of each examined technique.

5) “4. Challenges and Future Perspectives” and “5. Conclusion” (lines 527-582). These sections perfectly remark the most relevant outcomes found by the authors in this field and the promising future prospectives. It may be also opportune to highlight the potential future action lines to pursue the topic covered in this work.

Round 2

Reviewer 2 Report

Comments and Suggestions for Authors

The authors made extensive changes to the original manuscript to address the suggestions and requests of the three reviewers. They did an excellent job. I am truly satisfied with the revised version of the manuscript, which is now ready for publication.